# Water-Repellent Fluoropolymer-Based Coatings

**Guillermo Paz-Gómez [1], Juan Carlos del Caño-Ochoa [2], Oscar Rodríguez-Alabanda [2,*],
Pablo E. Romero [2], Miguel Cabrerizo-Vílchez [1], Guillermo Guerrero-Vaca [2]
and Miguel Angel Rodríguez-Valverde [1]**

[1]   Bio-colloid and Fluid Physics Group, Applied Physics Department, Faculty of Sciences,
University of Granada, Campus de Fuentenueva s/n, ES 18071 Granada, Spain; gpaz@ugr.es (G.P.-G.);
mcabre@ugr.es (M.C.-V.); marodri@ugr.es (M.A.R.-V.)

[2]   Department of Mechanical Engineering, University of Cordoba, Campus de Rabanales,
Ctra. Madrid-Cádiz km 396, ES14071 Córdoba, Spain; jcoochoa_069@hotmail.es (J.C.d.C.-O.);
p62rocap@uco.es (P.E.R.); guillermo.guerrero@uco.es (G.G.-V.)

*   Correspondence: orodriguez@uco.es

**Abstract:** Fluoropolymer-based coatings are widely used for release applications. However, these hydrophobic surfaces do not reveal a significantly low adhesion. Water repellency incorporated to fluoropolymer coatings might enhance their release performance. In this work, we focused on the surface texturing of a well-known polytetrafluoroethylene (PTFE)-based coating. We explored as texturing routes: sanding, sandblasting and laser ablation. We examined the surface roughness with white light confocal microscopy and the surface morphology with environmental scanning electron microscopy (ESEM). Water-repellent fluoropolymer coatings were reproduced in all cases, although with different degree, parametrized with bounces of water drops (4–5 μL). Laser ablation enabled the lowest adhesion of coatings with 24 ± 2 bounces. This result and the current development of laser patterning for industry assure the incipient use of laser ablation for release coatings.

**Keywords:** release coatings; water repellency; fluoropolymer; polytetrafluoroethylene; laser ablation

## 1. Introduction

Release/demolding coatings based on fluoropolymers or fluoro-silicones are widely used in automotive and footwear manufacturing [1], as well as the cookware and food industries [2–5] among others. However, although the chemical formulation of the coatings is well-established, there are some challenges still to resolve such as durability and optimal demolding. In other respects, liquid-repellent materials enable the removal of liquid from their surface. A solid surface is referred to as superhydrophobic [6] when its contact angle is higher than 150° and its contact angle hysteresis is lower than 10° (low shear adhesion). These low-adhesion surfaces reveal different properties such as self-cleaning [7,8] or anti-icing [9,10]. We postulate that water repellency might enhance the release performance of fluoropolymer coatings [11].

Some previous works demonstrated that both the polytetrafluoroethylene (PTFE) particle size [10] and the morphology of the fluoropolymer contact surface have a direct influence on the water repellency properties offered by the coating. In the case of PTFE coatings, a microscopic surface morphology in which spherical nanoparticles have been produced is able to present contact angle values greater than 140° [12]. Two distinct hypotheses are classically proposed to explain this effect. On one hand, this rough pattern increases the surface contact area with the water drop, and on the other hand air remains trapped in micro-chambers below the drop, favoring superhydrophobic behavior [13,14].

Many techniques and processes can be applied to obtain superhydrophobia on PTFE coatings: PTFE film extension [15] consists in the mechanical alteration of the density of the fibrous crystals by axial extension to create polytetrafluoroethylene superhydrophobic surfaces; metallic templates [16] or simple filter paper templates [17] can be used to obtain a superhydrophobic "lotus-leaf" structure on the PTFE coating surface, the template is simply printed on the coating film when it is still soft and is removed just after the final sintering process. Furthermore, cold pressing of the PTFE film [18] or hot stamping [19] are other possible ways. The introduction of a micro- and nanoscale roughness onto the surface of a low surface energy material as PTFE is the best option to get superhydrophobic surfaces. This "hierarchical" surface morphology can be reproduced by different physical or technological procedures: the emulsion of metallic particles with PTFE [20,21], PTFE coating surface treatment with plasma [22–29] or Nd:YAG laser ablation, among other laser technologies [30–37], are an example of this. Other research work has addressed metals, which are roughened and then chemically treated to decrease their surface energy [38], or incorporating directly surface roughness or micro-texture on a hydrophobic material [39–41].

In the present work, water-repellent fluoropolymer coatings were reproduced by three different procedures: polishing, sandblasting and Nd:YAG (CW) laser ablation. The water-repellency level has been measured by the waterdrop bouncing test [42,43]. Unlike other previous works consulted in which this experimental test has been used to analyze the morphology of the droplet during the cycle of one or several bounces [43], the number of waterdrop bounces on the surface under study was counted to measure the degree of water-repellency in each treated PTFE coating sample. The results of this experimental research showed that Nd:YAG laser ablation enabled much lower adhesion of coatings, in terms of the number of bounces reproduced by the waterdrop onto the coating surface, compared with the adhesion observed in the samples treated by polishing and/or sandblasting. The results of this work demonstrated that Nd:YAG laser is a technique to be considered for the industrial treatment of PTFE surfaces aimed at applications that require the maximum degree of water-repellency feature.

As has been mentioned, there are different techniques that may be applied to provide surface texture to a hydrophobic coating. Sandpaper polishing (sanding) [44] and sandblasting [45] are well-known routes with direct scaling-up. Laser ablation is also used to modify the surface of coated materials [30,46]. The conditions of ablation may be largely varied, and it further allows the surface patterning.

In this work, we conducted a systematic study on superhydrophobic (SH) fluoropolymer coatings in the range of water-repellency regime. The coatings were engineered, separately, by sandpaper polishing, sandblasting and laser ablation. We measured the final roughness of the SH coatings with a white-light confocal microscope and their surface morphology with an ESEM microscope. We characterized their water adhesion [42] with the "bouncing drop" experiment. Laser ablation points out to be the best strategy to produce water-repellent fluoropolymer coatings.

## 2. Materials and Methods

We selected for this work the fluoropolymer ETERNA©, kindly provided by the company Whitford (Whitford Company, Runcorn, UK). This product is a commercial composite of polytetrafluoroethylene (PTFE), widely used in industry. The polymer was deposited on Al–Mg substrates (EN-AW 5754) and then cured in an oven. The coating is applied in multilayer. The layers are applied by spraying aqueous dispersions of PTFE onto the substrates. A HVLP type gun (high volume and low pressure)(Aerometal, Barcelona, Spain), adjusted at 0.3 MPa, was used for this purpose. The base layer was obtained after the application of Xylan XLR 17-080 (Whitford Corporation, Elverson, PA, USA) and was dried at 100 °C for 5 min, reaching a thickness of 8–10 μm. The second layer is Xylan XLR 17-353/D9172 Emerald Green. The multilayer coating reaches 20–25 μm thickness after curing.

Commercial release coatings are typically modified by adding inorganic fillers [47]. The fillers are mixed in the aqueous dispersions of PTFE and they remain fixed to the coating once cured. The fillers intend to improve the resistance to abrasion and wear for long-lasting coatings. Fillers alter the

polymer properties at high temperatures and in the case of the filler doping of PTFE, the hardness is typically increased [47].

The temperature of the coating annealing was varied accordingly from 350 °C up to 440 °C, above the melting point of PTFE (342 °C). The annealed PTFE-based coatings were prepared by the company Tecnimacor, S.L. (Córdoba, Spain).

### 2.1. Polishing

Several SiC sandpapers with different grit numbers were used. The grit numbers employed were P320, P500, P800 and P1000, according to the standard ISO [48]. The polishing process was performed manually and with a polishing machine. For the manual polishing, the samples were abraded against each sandpaper for 10 cycles. The automatic polishing was performed with a Struers TegraPOL-11 industrial polishing machine (Struers, Madrid, Spain). The coating and the polishing disc were both rotate clockwise for 5 s, with a loading mass of 90 g equivalent to a pressure of 2.2 kPa.

After polishing, the coatings were cleaned with Micro 90© solution, ethanol 96% and generously rinsed with distilled water.

### 2.2. Sandblasting

Unlike the polishing technique, in this section the curing temperature was fixed to 410 °C. We conducted sandblasting process with a Sandblast Cabinet, model CAT990 (Aslak S.L, San Quirze del Valles, Barcelona, Spain). A standardized blasting on the target surface is important for obtaining statistically significant results. A gun with a nozzle of 6 mm diameter was placed at 30 cm distance to the sample and inclined 45° with respect to the surface. The relevant abrasion process parameters in commercial sanding machines are basically the contact pressure and the peripheral speed. The operating parameters of the experiments performed in this work should be similar to an industrial sanding process but are limited by the equipment employed to sand the coatings. Three different corundums were used with different mean grain size and hardness: Armex Maintenance Formula XL bicarbonate (270 μm), WSK Gr. 46 Alumina white corundum (340 μm) and RBT9 Gr. 90 Alumina brown corundum (150 μm).

The bicarbonate abrasive has a hardness value of 2.5 in the Mohs scale, and both the white and brown corundum, close to 9. With each corundum, different values for air pressure and blasting time were explored. The air pressure was varied between 0.1 and 0.3 MPa, and the blasting time between 10 and 40 s. The sandblasted coatings were then intensively cleaned with Micro90© solution, ethanol 96% and distilled water.

### 2.3. Laser Ablation

The laser ablation process was performed with a pulsed/continuous wave laser model Rofin PowerLine E-20 SHG II (Rofin-Baasel España, S.L. Unip., Navarra, Spain). This type of laser technology can work both in pulsed mode and in continuous wavelength mode and is based in a Nd:YAG solid-state laser. The wavelength was 1064 nm.

The pulse frequency of the laser beam was set to 15 kHz, in continuous wavelength mode, and the width of the laser spot was adjusted to 10–15 μm. The beam velocity was varied from 50 mm/s up to 1000 mm/s and, for each one, different fluence values were tested. The pattern ablated on the coatings was always a grid of parallel lines, tilted 45° and separated 20 μm, over an area of 20 × 20 mm$^2$. The ablated coatings were cleaned with Micro90© solution, ethanol 96% and distilled water.

### 2.4. Surface Topography and Morphology Analysis

Surface roughness was measured with a white light confocal microscope Sensofar, model Plμ 2300 (Sensofar, Barcelona, Spain). The magnification used was 50×, the scanning area was 285.38 × 209.62 μm$^2$ and the Z-resolution was 0.2 μm. Four topographies were acquired for each sample, and the roughness parameters used were the arithmetic and root mean square roughness, $R_a$ and $R_q$, respectively.

The engineered coatings were also studied with an Environmental Scanning Electron Microscopy (ESEM) FEI, QuemScan650F model (Thermo Fischer Scientific, Eugene, OR, USA), using a voltage of 3 kV. The detection mode was set for scattered secondary electrons. The same microscope was employed to perform energy-dispersive X-ray spectroscopy (EDX) to explore the surface chemical composition of the coatings. The detector in this case used back-scattered electrons, and the voltage of the beam was 5 kV.

The three-dimensional morphology of the polishing and sandblasting treated coatings has been analyzed by an optical surface profiler, model Plµ 2300 by Sensofar (Sensofar, Barcelona, Spain) and the same analysis has been made for Nd:YAG laser ablated samples using a digital microscope, model DVM6 by Leica (Leica Microsystems, Hospitalet de Llobregat, Spain).

## 2.5. Bouncing Drop

For superhydrophobic surfaces, the sessile drop method provides values of contact angle or critical sliding angle with low resolution. The difficult localization of the contact points of non-wetting drops, the limitations of the conventional optical devices and numerical fitting of drop profiles with high contact angles [49], the tolerance of standard inclinometers working at very low tilt angles (<5°) and the monitoring of "restless" drops placed on water-repellent surfaces [50] require establishing new methodologies.

On the other hand, methods based on rapidly moving drops avoid the potential complications in measuring static/quasi-static water contact angles on superhydrophobic surfaces. Water bouncing can be used to determine the hydrophobicity of a surface [42], with a correlation between water contact angle and number of bounces. Extensive testing demonstrated that surfaces with a rounded microstructure demonstrate water bouncing when water contact angles are greater than 151°, however examination of surfaces that promote water pinning showed that water contact angles had to exceed 156° for water drops to bounce.

The bouncing drop experiments were employed to provide information about the dynamic adhesion of the coatings [38]. This adhesion, referred to as "tensile adhesion", is parametrized by the number of bounces of a water drop. This parameter intends to connect water repellency with non-stick functionality [11]. The term tensile adhesion refers to the opposition of the surface to allow the removal of liquid by forces that act vertically to the surface. We focus on this property because the goal is to improve the PTFE coatings for the use in demolding applications. In this work, we counted the bounces of a water drop of $4.2 \pm 0.2$ µL, released from a height of $10.1 \pm 0.2$ mm. Three water drops were released for each sample. The drop was formed with a needle of $0.25 \pm 0.05$ mm diameter. The sequence of bounces was recorded with a high-speed camera, Phantom MIRO 4 Monochrome (Vision Research Ltd. U.K., Bedford, UK), at 2800 frames per second (fps), the image resolution was 400 × 504 pixels and the exposure time was 235 µs.

## 2.6. Sliding Drop

To measure the critical sliding/roll-off angle of sessile drops (lateral adhesion) on the modified coatings, we used the tilting plate method. We typically used 60 µL drops of Milli-Q water. We selected this drop volume to increase the density of metastable drop configurations separated by smaller energy barriers and further the spatial resolution of the method. Drops were gently deposited at the center of the sample, which is fixed to the tilting platform, oriented horizontally. The drop placement was non-trivial because the drops rolled on the samples very easily. Once the drop was deposited and static, the platform was automatically inclined at a constant rate (5°/s). The inclination angle of the platform ranged from 0° to 60° (±0.01°). We registered the inclination angle in terms of time. Side views of the drop were captured simultaneously at 16 fps. We monitored the onset of the contact line motion at both sides (upside or downside) of the profile of the inclined drop.

## 3. Results and Discussion

### 3.1. Annealed Coatings

The EDX results of the PTFE-based coatings are shown in Table 1. None trace of unexpected species was found. The C and F signals validate the presence of a thick fluoropolymer coating on the alloy metallic substrate. EDX does not provide information on the polymer crosslinking as the annealing process produced at each curing temperature.

**Table 1.** Chemical composition (in %) of the fluoropolymer coatings.

| C | F | Si | Al | Fe |
|---|---|----|----|----|
| 24.72 | 75 | 0.14 | 0.08 | 0.06 |

In Figure 1, we show the effect of the annealing on the surface roughness of the final PTFE coatings prepared at different temperatures. Above 365 °C, the PTFE melting point, the surface roughness increases due to the cross-linking of the fluoropolymer (lower crystallinity degree).

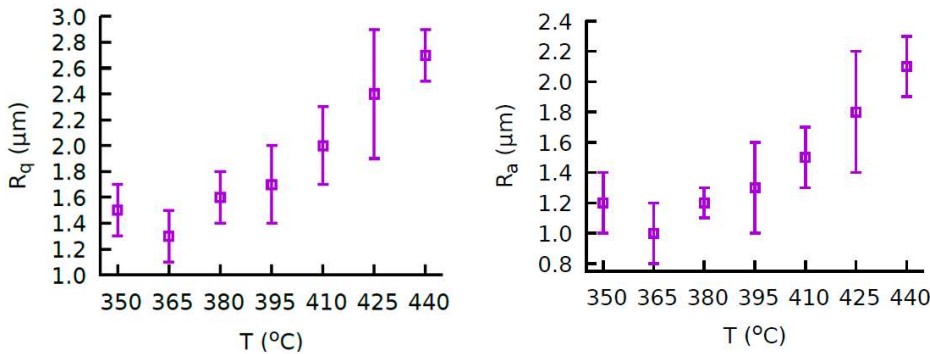

**Figure 1.** Root-mean square and arithmetic roughness of the fluoropolymer coatings cured at different temperatures. Above the melting point, the roughness amplitude increases as the temperature.

In Table 2, we compile the hardness values of the cured coatings, provided by the pencil test standard [51]. From Figure 1 and Table 2, the temperature 365 °C reveals a minimum point where the trend of both roughness and hardness values changed. Above this temperature, up to 440 °C, both the surface roughness and hardness increase.

**Table 2.** Hardness of the PTFE coatings, without texturization, at different curing temperatures (pencil hardness standard decreasing scale: 6H> 5H> 4H> 3H> 2H> H> F> HB> B> 2B> 3B> 4B> 5B).

| Temperature (°C) | Hardness |
|------------------|----------|
| 350 | HB |
| 365 | HB-B |
| 380 | F |
| 395 | F |
| 410 | H |
| 425 | H |
| 440 | F |

In the images of the Figure 2 shows a summary of the different technics used. The Figure 2a presents the as-received coatings revealed a fibrous structure like a tangle, formed during the curing.

The processes developed in the polymer bulk for annealing are crucial to establish the final performance of the coating. Above 360 °C, the PTFE loses molecular weight, which promotes the crystallization of the polymer, hardening it. The greater the crystallization degree, greater the surface roughness. The filler doping of PTFE increases its hardness [47].

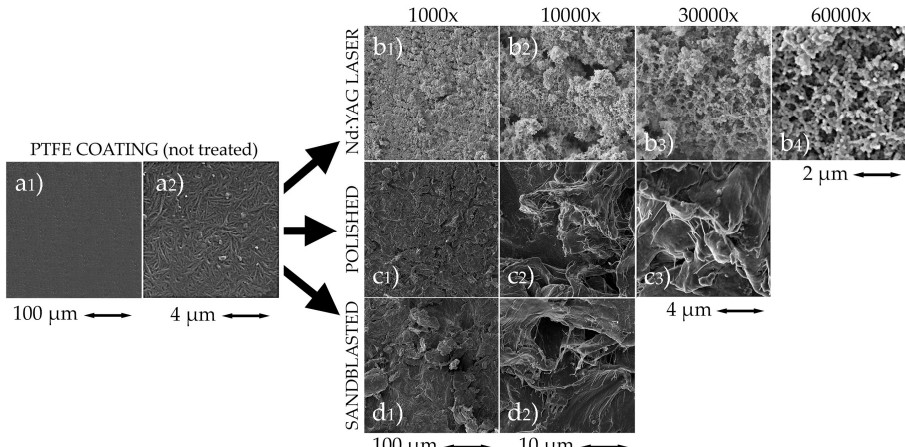

**Figure 2.** ESEM images of the fluoropolymer coatings after each surface treatment: (**a**) as-received coating showing the small size of the fibers; (**b**) laser ablated coating (fluence 260.4 J/cm$^2$ and beam speed of 100 mm/s); (**c**) polished coating (P500 grit cured at 395 °C); and (**d**) sandblasted coating (brown corundum ejected at 0.1 MPa for 40 s).

As seen in Figure 3, their water shear adhesion, parametrized with the sliding angle (SA) of water drops, was not sufficiently low because the SA is always greater than 10°.

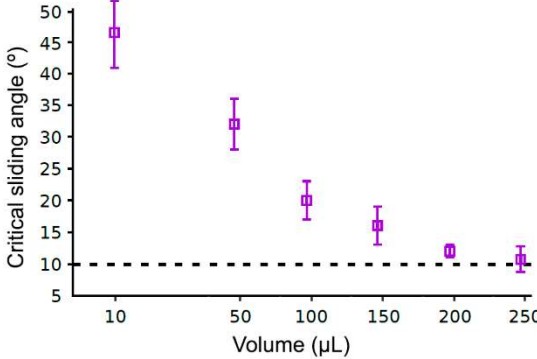

**Figure 3.** Sliding Angle in a PTFE coating cured at 410 °C (without further surface treatment), for different drop volumes of Milli-Q water (for 5 μL, we measured SA ≥ 60°).

The water contact angle of the as-received fluoropolymer coatings was greater than 90°, for any curing temperature, but they were not water-repellent. It is well-established that water-repellent surfaces are identified as non-wetting surfaces with SA values much lower than 10° [52].

### 3.2. Polished Coatings

In Figure 2d, the sanded coating looks like a "hairy" surface that may allow air retaining [53]. The fibers of the fluoropolymer coating raised up after polishing. Air flows readily through the channels below the water drop.

In Table 3, we collect the values of SA of the most representative sanded coatings. The SA values were always ≤ 2°, even the accelerometer fixed to the tilting platform was unable to resolve with positive tilts the drop motion (SA < 0.01°). This behavior was also found for the rest of coatings (not shown). In terms of tensile adhesion, we discarded the polished coatings with none drop bounce onto them. The best samples are presented in Table 3. A number of bounces lower than 1 reveals different

runs on a coating where rarely the drop rebounded. It is worth mentioning that no bounce was measured in those coatings cured at excessively high temperature (greater than 410 °C). An excess of curing can produce significant alterations in the chemical structure of the polymer coating.

**Table 3.** Surface roughness ($R_a$), Critical Sliding Angle (SA) and drop bounces for the sanded fluoropolymer coatings cured at different temperatures. Number of bounces > 3 are highlighted in bold. The standard deviation reflects the data scattering.

| Grit/Sintering Temperature (°C) | $R_a$ (μm) | SA (°) | Bounces |
|---|---|---|---|
| P320/380 | 2.6 ± 0.4 | ≤2° | 0.2 ± 0.3 |
| P320/410 | 6.9 ± 1.8 | ≤2° | 0.1 ± 0.3 |
| P500/350 | 5.1 ± 1.5 | ≤2° | 0.6 ± 0.8 |
| P500/365 | 3.7 ± 0.6 | <0.01° | 5.7 ± 0.6 |
| P500/380 | 3.1 ± 0.3 | <0.01° | 3.3 ± 0.6 |
| P500/395 | 4.9 ± 0.1 | <0.01° | 4.0 ± 1.0 |
| P500/410 | 4.2 ± 0.7 | <0.01° | 2.1 ± 1.9 |
| P500/425 | 2.6 ± 0.1 | <0.01° | 2.2 ± 2.6 |
| P800/380 | 2.1 ± 0.6 | <0.01° | 2.1 ± 0.2 |
| P800/410 | 2.3 ± 0.3 | <0.01° | 4.7 ± 0.6 |

The combined effect of polymer cross-linking, during the curing, and polishing provided a particular morphology and roughness that led to water repellency at different degree. This way, different combinations of curing temperature [18] and sandpaper grit number allowed to reach different values of water-repellency collected in Table 3.

The coatings polished by machine were, in all cases, less homogeneous than the hand-polished ones and with a lower number of bounces. The automatic polishing does not follow the surface deformations of the metallic substrate, leading to a non-uniform treatment of the surface.

*3.3. Sandblasted Coatings*

In Table 4 we compile the best cases in which the water tensile adhesion of the sandblasted coatings was significant. The roughness is greater than in the polished coatings, mainly because the mean grain size of the corundums is greater than the grain of the sandpapers used.

Figure 2d illustrates a surface rougher than the polished coatings (Figure 2c). With this structure, the air trapping is less effective, and the water-repellent behavior is reduced.

**Table 4.** Surface roughness ($R_a$) and drop bounces for the sandblasted fluoropolymer coatings with lower tensile adhesion. In the first column, we show the corundum used, the blasting time and pressure.

| Corundum, Time (s)/Pressure (MPa) | $R_a$ (μm) | Bounces |
|---|---|---|
| Bicarbonate, 25/0.3 | 6.5 ± 0.9 | 3.7 ± 1.5 |
| Bicarbonate, 15/0.3 | 8.0 ± 2.0 | 3.0 ± 0.0 |
| Brown corundum, 40/0.1 | 7.0 ± 2.0 | 4.0 ± 1.7 |

Sandblasting produces roughness by the aggressive erosion of the surface. Due to this, special care was taken to avoid the damage or removal of the coating. The EDX spectroscopy (not shown) revealed a greater presence of alumina in the sand-blasted coating, which confirms that part of the corundum remained within the coating and was not removed for the cleaning process.

By using two abrasives and different values of pressure and blasting time, we approximately reproduced the same values of bounces. With the softest corundum (bicarbonate), we needed greater pressure and time to reach a uniform texture on the coating.

### 3.4. Laser Ablated Coatings

In Figure 4 we show the bounce values for the laser-ablated coatings for various regimes of laser beam fluence. With this surface treatment, we found the highest values of water tensile adhesion. Every tested velocity range produces a maximum of bounces that corresponds to a certain value of fluence of the laser beam, being the highest values in terms of water-repellence those obtained on the surfaces treated at 100 mm/s. Then, as the beam speed increases, the tensile adhesion peak decreases.

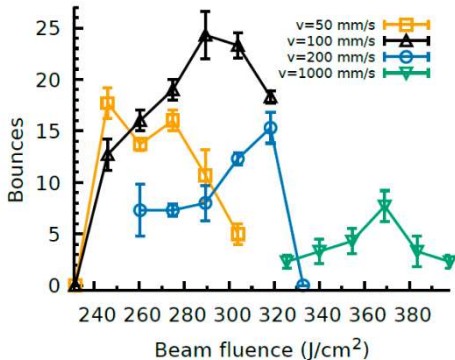

**Figure 4.** Drop bounces for the coatings ablated with different values of beam velocity and fluence.

The main advantage of laser patterning is the structured texturization rather than random, like happens with sandblasting and sanding. Laser ablation allows to create channels with a well-defined geometry, where air can readily be entrapped.

Figure 2b shows a laser ablated coating ($\phi_0$ =260.4 J/cm$^2$, $v$ = 100 mm/s). This surface looks smoother and it reveals a great number of well-shaped fibrous channels. With this surface, we measured 16 ± 1 bounces. The surface roughness was measured for samples ablated with different fluences and beam velocities and the $R_a$-values are within the range 3.6 ± 0.9–5.3 ± 0.5 µm.

In the SEM images shown in Figure 5a,b the textures obtained by polishing and sandblasting present a very marked heterogeneity and the hierarchical characteristic in its micro and nano structure is disordered, the pattern is based in amorphous protrusions in both cases. In addition, these protuberances have sharp edges, unlike the smoothness of the rounded protrusions obtained in the case of the laser-treated surface. Both, polished and sandblasted patterns, are relatively homogeneous in the micro-scale (1000× magnification) but heterogeneous and amorphous in the nano-scale level.

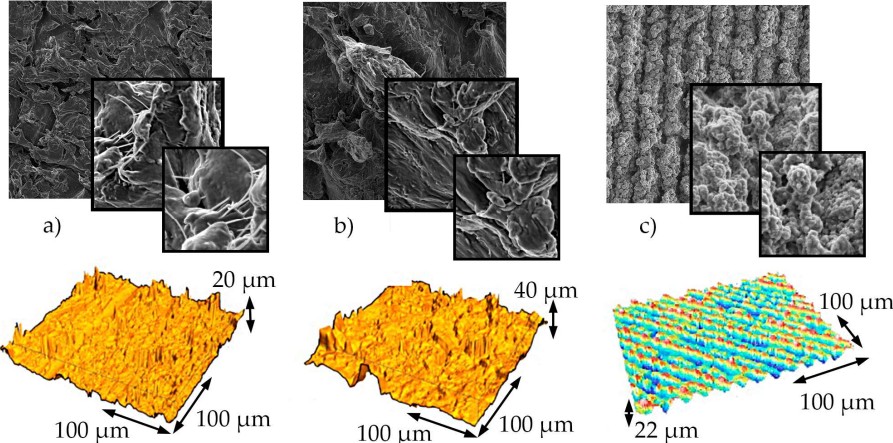

**Figure 5.** AFM and SEM images showing the morphology on the PTFE coating surface treated by: (**a**) polishing with P500 and cured at 380 °C, (**b**) sandblasting with brown corundum, (**c**) Nd:YAG laser ablation at $\phi_0$ = 260.4 J/cm$^2$, $v$ = 200 mm/s (SEM magnification ranges: big size 1000×, medium size 30000x, small size 60000×).

In the SEM images of Figure 5c, the surface patterning obtained with Nd:YAG laser ablation reveals soft-rounded shapes in a well equilibrated distribution in dual scale. At micro-scale, soft rounded shapes are distributed in parallel stripes originated by the scanning movement of the laser beam. At different levels of the nano-scale (30,000×, 60,000×), it is appreciated a second pattern of soft-rounded protrusions distributed on the protrusions of greater size. The surface pattern is repeated at micro and nano scale, constituting a hierarchical morphology.

This characteristic feature is what causes that the PTFE coatings treated with Nd: YAG laser lead a weaker water-surface interaction and, therefore, a higher level of water-repellency than those PTFE surfaces treated with polishing or sandblasting procedures. The experiment with Nd:YAG laser allows an observation of the effect of both parameters, fluence and beam velocity, in water-repellency.

The images in the Figure 6 evidences the effect of the beam velocity applying the same level of energy per surface unit ($\phi_0$ =260.4 J/cm$^2$). A higher velocity (200 mm/s) produces more pronounced channels in the micro-level, while at the optimum speed experienced (100 mm/s) the ridges practically melted with each other, creating a smoother micro-texture. At the nano-level, the morphology shows spheroidal particles much more defined and smaller than those obtained in the specimen processed at higher velocity range.

The (a) specimen reached 16 ± 1 bounces while the (b) specimen reached 7 ± 1 bounces in the bouncing drop test.

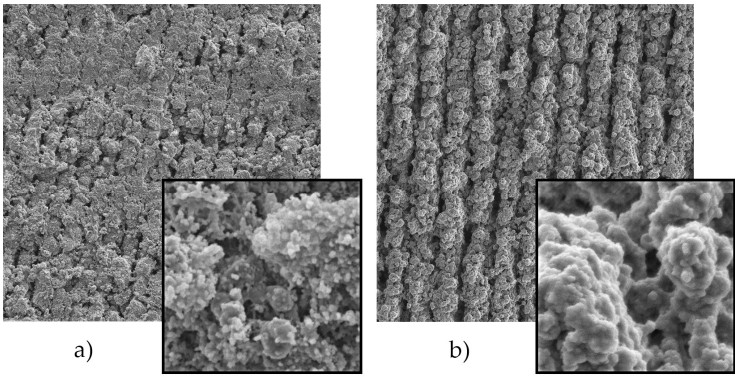

a)　　　　　　　　　b)

**Figure 6.** SEM images showing the morphology on the PTFE coating surface treated by Nd:YAG laser ablation at $\phi_0$ = 260 J/cm$^2$: (**a**) $v$ =100 mm/s, (**b**) $v$ = 200 mm/s (SEM magnification ranges: big size 1000×, small size 30000×).

The surface roughness was measured for laser ablated specimens shown in the SEM images of the Figure 6; Figure 7, corresponding the $R_a$-values in the Table 5.

**Table 5.** Surface roughness ($R_a$) and drop bounces for the Nd:YAG laser ablated PTFE coatings processed at the optimum ranges of fluence and velocity. Comparison with the optimum results obtained by other methods.

| Method | Fluence (J/cm$^2$) | Beam Velocity (mm/s) | $R_a$ (μm) | Bounces |
|---|---|---|---|---|
| Polished (P500, 380 °C) | – | – | 7.0 ± 2.0 | 4.0 ± 1.7 |
| Sandblasted (brown corundum) | – | – | 3.1 ± 0.3 | 3.3 ± 0.6 |
| Nd:YAG Laser | 260 | 200 | 3.9 ± 0.9 | 7.0 ± 2.5 |
| Nd:YAG Laser | 260 | 100 | 4.0 ± 0.2 | 16.0 ± 1.0 |
| Nd:YAG Laser | 290 | 100 | 4.7 ± 0.4 | 24.0 ± 2.0 |

The best result produced by Nd:YAG laser ablation, in terms of water-repellency, has reached up to 24 ± 2 bounces in the bouncing drop test and the morphology on the treated PTFE coating surface can be observed in the images obtained through SEM and presented in Figure 7.

a)　　　　　　　　　　b)　　　　　　　　　　c)

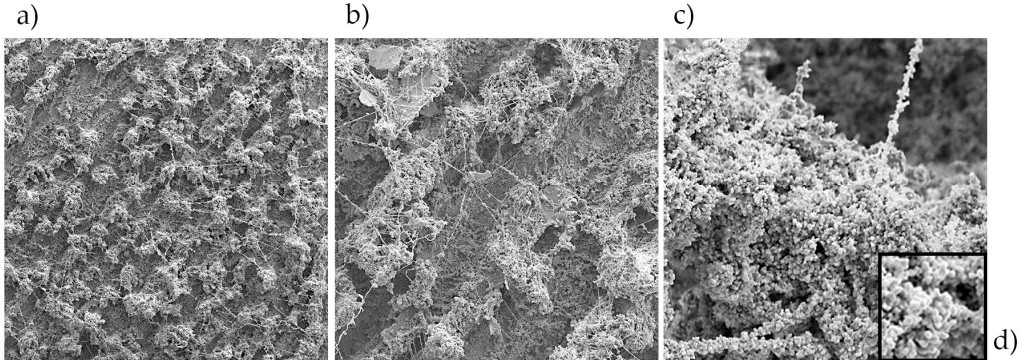

d)

**Figure 7.** SEM images showing the morphology on the PTFE coating surface treated by Nd:YAG laser ablation at $\phi_0$ = 290 J/cm², $v$ = 100 mm/s at magnification ranges: (**a**) 150×; (**b**) 500×; (**c**) 10,000×; (**d**) 15,000×.

In this coating, the surface morphology is constituted by rounded protrusions combined with fibrous micro-structures and distributed homogeneously over the treated surface. At nano-scale (10000×–15000×), it is possible to observe a fibrous nano-structures formed by very small soft-rounded and spheroidal shapes on the surface of the protrusions of the previously defined micro-pattern. These soft-rounded forms, combined with the fibrous nano-structures, reveal a pattern that constitutes a hierarchical morphology.

## 4. Conclusions

We prepared water-repellent fluoropolymer coatings by roughening a commercial release coating. We selected sandpaper polishing, sandblasting and Nd:YAG (CW) ablation, due to their industry scale-up, to change the surface morphology and roughness of the hydrophobic coating. In all cases, we found certain conditions that led to water-repellency phenomena.

- The sliding/roll-off angle measurements become deficient by itself for the characterization of water repellency level in the case of the polished PTFE coatings and the rest of treated coatings. Bouncing drop test is a suitable alternative.
- Laser ablation was the best route to achieve high water-repelling properties. For a laser beam fluence of 289.4 J/cm² and velocity of 100 mm/s, the bouncing drop reproduced up to 24 ± 2 bounces.
- Laser ablation achieved a homogeneous pattern in terms of micro-roughness, showing a hierarchical surface morphology with micro-scale and nano-scale round-shaped protrusions, more or less defined according to the energy and velocity applied in the laser ablation process.
- The polished and sandblasted PTFE coatings also show acceptable water-repelling properties.

Although the results of this experimental study have been satisfactory, further work should be addressed to determine the realistic response of modified fluoropolymer coatings at industrial scale. The durability should be studied to validate the potential application to industrial processes.

**Author Contributions:** Conceptualization, M.A.R.-V. and G.G.-V.; Methodology, M.A.R.-V., G.G.-V.; G.P.-G. and J.C.d.C.-O.; Validation, All authors; Formal Analysis, All authors; Resources, G.G.-V., M.A.R.-V. and M.C.-V.; Data acquisition, G.P.-G., J.C.d.C.-O , P.E.R. and O.R.-A.; Writing-Original Draft Preparation, G.P.-G. and M.A.R.-V.; Writing-Review & Editing, G.G.-V. and O.R.-A.

**Funding:** The research was funded by the projects MAT2014-60615-R and MAT2017-82182-R, funded by the Spanish Ministry of Economy and Competitiveness (MINECO-FEDER).

**Acknowledgments:** We would like to thank to the company TECNIMACOR S.L. (Córdoba, Spain) for the preparation of the coatings studied in this work.

**Conflicts of Interest:** The authors declare no conflict of interest.

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
