# Peer review of "Water-Repellent Fluoropolymer-Based Coatings"

_coatings, doi:10.3390/coatings9050293_

Reviewer 1 Report

The draft is well structured; the experimental design, results and conclusions are scientifically sound. In addition, the interest from a technical point of view is high.

Nevertheless, provided references regarding laser surface modification of polymers are not enough according the importance of Nd:YAG laser treatment contribution in the work. Important background on polymer surface modification by 1064 nm laser wavelength can be found in:

A. Riveiro et al. Laser Surface Texturing of Polymers for Biomedical Applications, Front. Phys., 27 February 2018 | https://doi.org/10.3389/fphy.2018.00016

Aránzazu del Campo et al. Fabrication Approaches for Generating Complex Micro- and Nanopatterns on Polymeric Surfaces. Chem. Rev., 2008, 108 (3), pp 911–945 DOI: 10.1021/cr050018y

Lippert T. Laser application of polymers. Adv Polym Sci. (2004) 168:51–246. doi: 10.1007/b12682

Lippert T. Interaction of photons with polymers: from surface modification to ablation. Plasma Process Polym. (2005) 2:525–46. doi: 10.1002/ppap.200500036

The new paragraphs require slight language grammar edition.

2 Materials and methods:

Authors must specify very clearly the pulse repetition rate (¿15 KHz?) and pulse duration time (ns? ms? range), particularly if laser fluence parameter is employed to discriminate laser processing conditions. Moreover, the laser beam quality factor and the used optics to reach 1 micron spot size must be detailed, and the way to measure such spot size (a higher spot size would be expected in case of F-Theta Lens for 1064 nm, and higher spot size is suggested by the SEM/confocal microscope observations –Figures 2d and 6c-).

3 Results

Figure 2a is suggested to be replaced by a similar magnification micrograph to Figure 2b-d for comparison purposes.

The parameter to represent observed surface roughness must be unified (Rq and Ra are used, it is suggested to show Ra values in the graph of Figure 1).

Author Response

Dear Reviewer,

First of all, we want to thank the Reviewer for the time and work dedicated to reviewing our work.

Mindfully, we will be very grateful for any appreciation to improve the final version of our work.

Thanks, and regards.

Reviewer 2 Report

The paper describes the possibility to obtain water-repellent fluoropolymer coatings by three different procedures: polishing, sandblasting and

Nd:YAG continuous wavelength (CW) laser ablation; their water-repellency level has been measured by the waterdrop bouncing test

counting the number of waterdrop bounces on the surface.

The results showed that Nd:YAG (CW) laser ablation enabled much lower adhesion of coatings, in terms of the number of bounces reproduced

by the waterdrop onto the coating surface, compared with the adhesion observed in the samples treated by polishing and/or sandblasting.

The analyses support very well the conclusions.

The English edition of the manuscript is readable and understandable.

The results are proper for publishing in this journal and are well suited to Coatings.

I noticed that the manuscript was reviewed beforehand and I found that the form received was well prepared. I would suggest publishing the paper

in this form.

With respect,

Author Response

Dear Reviewer,

First of all, we want to thank the Reviewer for the time and work dedicated to reviewing our work. We want to inform you that some improvements were done after this first round of revisions and invite to revise the new version of the manuscript.

We will be very grateful for any appreciation to improve the final version of our work.

Thanks, and regards.

Reviewer 3 Report

The paper by Gómez et al. presents some interesting results, yet further experiments are required before it can be accepted. I recommend major revision. My specific comments are attached below.

1)    Literature should be improved. I know that a lot of work has been done with laser micromachining. 

Two examples:

Eur. Phys. J. Appl. Phys. 25, 33–38 (2004) DOI: 10.1051/epjap:2003083

Int J Mol Sci. 2014 Aug; 15(8): 13681–13696. doi: 10.3390/ijms150813681

With many more existing.

Additionally, add references about other texturing techniques, such as plasma.

Several works on plasma texturing using plasma etching exist. 

2)    The bouncing test is interesting, however it is not acceptable without water static contact angle and hysteresis measurements. The authors should incorporate such measurements and then compare these data with the bouncing test for each texturing method they used. 

3)    New SEM images with higher magnification are required to highlight the effect of each texturing process. Authors report micro and nanotexturing and the scale in three out of four SEM images is 100 μm. It is impossible to see nanofeatures using this magnification scale.

4)    Figure 3 the drop volumes are extremely high, test low volumes below 10 μL

5)    The AFM images do not provide significant information in the way they are used. If possible calculate roughness rms to quantify your statements.

6)    Conclusions: lines 276-278

“Laser ablation allowed a homogeneous pattern in terms of micro-roughness and nanoroughness scale, showing a hierarchical surface morphology with larger and smaller protrusions distributed in parallel channels generated by the laser beam displacement.”

No evidence of hierarchical morphology are presented.

7) Last but not least. See also my first comment. Authors should use previous works to enhance their novelty. 

Minor:

line 16 abstract “surface texturization” rephrase to ”surface texturing”

Author Response

Dear Reviewer,

First of all, we want to thank the Reviewer for the time and work dedicated to reviewing our work. 

We will be very grateful for any appreciation to improve the final version of our work.

Thanks, and regards.

Round  2

Reviewer 3 Report

The revised version of the work by Gómez et al. is improved, yet several of my comments are not answered and therefore i recommend another round of revision. My specific comments are attached below.

1) The works about plasma texturing are out of date, please add some more recent.

Some examples: 

Superamphiphobic polymeric surfaces sustaining ultrahigh impact pressures of aqueous high‐and low‐surface‐tension mixtures, tested with laser‐induced forward transfer of drops. Advanced Materials 27 (13), 2231-2235

Is There a Threshold in the Antibacterial Action of Superhydrophobic Surfaces?, ACS applied materials & interfaces 9 (45), 39781-39789

Easy plasma nano-texturing of PTFE surface: From pyramid to unusual spherules-on-pyramid features, Applied Surface Science

Sphere-on-cone microstructures on Teflon surface: Repulsive behavior against impacting water droplets, Materials & Design 92, 1052-1061

2) Report water static contact angle and hysteresis for each method used for texturing. It is not acceptable to speak about water repellent surfaces and present no or limited data about contact angle measurements. 

3)    Figure 3 the drop volumes are extremely high. Repeat the measurements with lower volumes below 10μl. 

Author Response

Dear Reviewer,

First of all, we want to thank you for the time and work dedicated to reviewing our work. In the pdf file, we attach the answers and explanations (in bold font) on the indicated aspects that have allowed to improve the article.

Round  3

Reviewer 3 Report

I don't agree with what is written in Sections 2.5 and 3.2, contact angle measurement is a well established method to study the macroscopic wetting behaviour of a surface and everyone working in surface science uses it. Authors in reference 55, don't say that we must stop measuring CA. They propose an additional method to observe adhesion through force measurements. My personal opinion is that the way reference 55 is used inside the text is misleading.

The surfaces reported in this work are not considered as highly repellent surfaces and i am not sure if the CA measurements are done properly. For example table 3 shows very small differences between ACA and RCA that should result in surfaces in which drops are bouncing yet this is not observed. 

I propose authors to take their time and improve their manuscript. To do so they need to have CA measurements, hysteresis or sliding angle measurements and their bouncing test for every surface they present. From my point of view, in this work different experiments are presented in each section that don't help the reader to understand differences. For example, in Sections 3.3 and 3.4 they don't report any CA.